# Avatar error in your favor: Embodied avatars can fix users' mistakes without them noticing

**Mathias Delahaye**[1,2]*, **Olaf Blanke**[2], **Ronan Boulic**[1], **Bruno Herbelin**[2]

**1** Immersive Interaction Research Group, École Polytechnique Fédérale de Lausanne (EPFL), Lausanne, Switzerland, **2** Laboratory of Cognitive Neuroscience, Brain Mind Institute, École Polytechnique Fédérale de Lausanne (EPFL), Lausanne, Switzerland

* mathias.delahaye@epfl.ch

## Abstract

In immersive Virtual Reality (VR), users can experience the subjective feeling of embodiment for the avatar representing them in a virtual world. This is known to be strongly supported by a high Sense of Agency (SoA) for the movements of the avatar that follows the user. In general, users do not self-attribute actions of their avatar that are different from the one they actually performed. The situation is less clear when actions of the avatar satisfies the intention of the user despite distortions and noticeable differences between user and avatar movements. Here, a within-subject experiment was condutected to determine wether a finger swap helping users to achieve a task would be more tolerated than one penalizing them. In particular, in a context of fast-paced finger movements and with clear correct or incorrect responses, we swapped the finger animation of the avatar (e.g. user moves the index finger, the avatar moves the middle one) to either automatically correct for spontaneous mistakes or to introduce incorrect responses. Subjects playing a VR game were asked to report when they noticed the introduction of a finger swap. Results based on 3256 trials (~24% of swaps noticed) show that swaps helping users have significantly fewer odds of being noticed (and with higher confidence) than the ones penalizing users. This demonstrates how the context and the intention for motor action are important factors for the SoA and for embodiment, opening new perspectives on how to design and study interactions in immersive VR.

## Introduction

We feel responsible for the actions we do and for the mistakes we make. In computer games, as in sport or at work, missing a button press under time pressure is self-attributed as a failure to succeed in the task. But if our avatar in Virtual Reality (VR) would automatically correct for it, pretending the mistake never occurred, would we still feel responsible for that mistake, or simply ignore it, or even not notice it at all?

The self-attribution of authorship for voluntary actions is defined as the Sense of Agency (SoA), which corresponds to the subjective feeling to be responsible for the action of our body [1–3]. The representation in VR of an avatar replicating a user's movements thanks to motion

**Data Availability Statement:** All relevant data are within the paper and its Supporting information files.

**Funding:** M.D. has been supported by Schweizerischer Nationalfonds zur Förderung der

Wissenschaftlichen Forschung (SNF) (https://snf.ch) with the grant 200020_178790. The funder had no role in study design, data collection and analysis, decision to publish, or preparation of the manuscript.

**Competing interests:** NO authors have competing interests.

capture is known to induce a strong SoA for the movement of the virtual body. This is illustrated through the adaptation from Slater et al. [4] of the "Rubber Hand Illusion" from Botvinick et al. [5] with the difference that the limb is not a physical limb, but a virtual one displayed using immersive technologies. Those immersive setups can also induce a sense of self-location in the Virtual Environment (VE) (i.e., I am located where the virtual body is located) which, carefully combined with the SoA can lead to the subjective experience of body ownership SoE (i.e., this virtual body is my body). This feeling of embodiment is beneficial for the user experience in VR, but any disruption of one of them can potentially lead to a break in embodiment [6]. Therefore, it is crucial not to disrupt the SoA to maintain the user's experience.

It is assumed that to perform this SoA judgment, an underlying neural process would compare a prediction of the sensory consequences of a movement with the actual feedback from our senses [7–10]. If both match, the SoA is high, but if a mismatch occurs, there is a loss of SoA. It was thus surprising to observe that humans could self-attribute actions that were distorted (i.e., When the user's movement differs from the avatar's motion) or performed by others [11, 12]. Prior results typically showed that, beyond a few hundred milliseconds of delay, people do not self-attribute the response [13, 14] therefore only real-time feedback are considered in this study.

In the seminal study of Nielsen et al., subjects were asked to draw a line while the experimenter secretly placed a mirror to replace the subject's real hand with someone else's hand doing the same task [11]. When both actions were synchronous, subjects experienced the alien limb (i.e., the limb that does not belong to the user) and its movements as their own. When the alien limb drew a curve instead of a line, subjects compensated for the error by making involuntary corrections in their movement while still considering the limb to be their own. With more advanced techniques using computer graphics, Burns et al. managed to introduce motion discrepancy in the middle of a movement and showed that users are much less sensitive to a visual-proprioceptive discrepancy (distortions) than to a visual artifact such as the interpenetration of the hand in an object [15]. Interestingly, informing participants of the possibility of a mismatch was shown to influence the tolerance to discrepancies. Burns' research hence showed that 45 degrees offset in the arm rotation could be unnoticed if the participant was not previously warned about the gradual introduction of this deformation, but only of around 18 degrees when the subject was informed [15]. Independently of such factors, this tolerance has been shown to be quite useful to progressively remove discrepancies between a real hand position and its virtual counterpart, such as for recovering a gap due to virtual constraints or to rub out tracking and animation imperfections [16].

Concerning finger movements, Krugwasser et al. observed that introducing angular distortions or temporal delays gradually reduce the SoA similarly to other effectors [17]. Importantly, they also report that spatial and temporal distortions affect less the SoA than anatomical distortions (the limb displayed moving is not the actual limb the user moves). They thus suggest that this higher sensitivity arises from the combination of a spatial discrepancy (the moving finger is not located where the displayed moving finger is) with an anatomical discrepancy, accumulating to a larger overall conflict.

To investigate anatomical distortion, Caspar et al. used a robotic hand placed on an over-raised wood plank [18]. The actual participant's hand was located just below, and both real and mechanical hands were placed above a physical keyboard. Through a mix of conditions where the motion was either congruent or incongruent (by transposing the movement of the index finger to the little finger), the authors evaluated participants' SoA. They observed that having a congruent mechanical hand leads to an SoA similar to the one experienced for the real hand while introducing the anatomical conflict significantly reduced it. They thus

concluded that matching the effector to achieve an outcome is a strong factor influencing the agency's judgment, as it links with the sense of embodiment for that effector.

Using a VR display apparatus, Salomon et al. further investigated the link between the swapping of finger motion and the impact on the self-attribution of the performed movement [19]. They also report the importance of embodiment for the judgment of SoA and, of primary importance for our question, they further showed that participants' accuracy was strongly affected by the movement they had viewed when asked to judge which movement they performed. This conflict elicits the possibility for self-attributing finger-swapped actions in VR.

In fact, the seminal work of Logan et al. previously demonstrated the possibility of a cognitive illusion of authorship, without VR, by asking skilled typists to type words on a computer while the visual feedback was either automatically corrected for typos or with inserted errors [12]. Their results show that typists typically took credit for correct output on the screen (i.e., interpreting corrected errors as their correct responses) and, more interestingly, that typists, who were unaware of the possible introduction of mistakes by the computer, also blamed themselves for inserted errors, considering the visual output resulted from their action.

Those observations seem to corroborate the idea that the cognitive illusion of authorship could be manipulated in VR such that participants would self-attribute a correction or a mistake introduced in VR. However, it is not known if the real-time feedback of the error (i.e., the participant does not wait for the end of the word to get the typed word displayed on the screen) would prevent such expectations from happening.

It has been shown that a continuous distortion introducing a spatial discrepancy between the real (hidden) and the virtual (visible) arm in a reaching task is rather well-tolerated [6, 20, 21]. More specifically, participants still report being the agent performing the action despite a relatively large distortion, typically when it helps them to reach a goal (around +2dB change in movement's speed in the study from Debarba et al. [20]) as opposed to when it prevents them from doing so. This tolerance for amplified or reduced movement cannot be interpreted solely as a limit in detection threshold, as it is influenced by other factors linked to the achievement of a task and to a more global Sense of Embodiment (SoE). Therefore, authors revealed that distortions helping users were more accepted than distortions hindering the movement, and that those distortions can be thus used to help (or penalize) users to reach their goals. It can thus be expected that even stronger distortions, such as changing the motion of a body part for another one, could also be tolerated although it is not yet proved.

Jeunet et al. evaluated three aspects of the SoA through fingers' animation manipulations as viewed in the first-person perspective in an HMD [22]. In this experiment, the authors manipulated the priority principle (i.e., the intention immediately precedes the action) by introducing temporal delays, the consistency principle (i.e., what is expected to be observed is observed) by swapping finger motion, and the exclusivity principle (i.e., one is the only apparent cause of the outcome) by randomly animating the hand. In line with former literature, they confirm a decrease in SoA when any manipulation was introduced, with the lowest agency score when consistency was altered (i.e., finger swaps). Interestingly, they also observe a correlation between the agency score and the level of immersion in VR, outlining the mutual interaction between immersion conditions and the level of SoA. However, these observations were made for isolated movements (not specific to fingers), independently of the execution of a goal-oriented task.

Overall, these studies show that introducing finger swaps reduces the SoA and that visual feedback tends to dominate over motor perception during these conflicts. However, it is unknown if, as for movement distortion in a reaching task, this would still apply to goal-oriented tasks. More specifically, it could be expected that helping or hindering the participant

would influence their SoA differently for finger-swapped actions (as is the case for Debarba et al.'s reaching study [20]).

Thus, the present study evaluates the impact of finger swaps during a goal-oriented situation through a challenging VR game in which participants have to validate buttons with fingers. More specifically, this paper analyzes whether participants would detect these anatomical swaps in two contexts: without and with spontaneous errors (SE). In the context without SE, we assess the condition of error introduction (**EI**: the subject does the right action but the swap prevents the user from validating the button), whereas, in the with SE context, we assess the condition of error correction (**EC**: the subject makes SE and the system corrects for them).

## Setup

The VR apparatus used for this experiment involves hardware and tangible objects as well as a representation of the user's avatar inside a 3D simulation running with Unity3D [23] (Fig 1a). To study finger swaps and to support the avatar's embodiment, the visible parts of the body are animated thanks to a Motion Capture (Mocap) system and an animation pipeline. As finger swaps must be introduced, the hands' animation pipeline is adapted to allow the permutation of fingers' motions as illustrated in Fig 2.

The Mocap system is a Phasespace Impulse X2 [24]. This tool converts markers (red LEDs) attached to the glove (Fig 1a) into 3D points in space. As optical Mocap is sensitive to visual occlusions, in particular for fingers tracking, we used the occlusion recovery process from Pavllo et al. [25]. For animating the avatar, marker positions are fed to analytical Inverse Kinematic (IK) algorithms. Lower body parts are not visible (under the table) and thus not animated. Participants are immersed in VR with an HTC Vive Pro Eye Head-Mounted Display (HMD) and see their avatar body in first person view.

Finally, to allow the user to report an event or validate steps, an *M-Audio SP-2* pedal (connected via an Arduino Uno [26] to the computer) is placed under the participants' foot. This system detects pedal press when the pedal reaches its mid-travel, triggering a falling edge detection on the microcontroller.

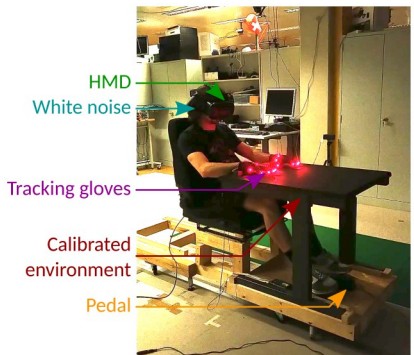 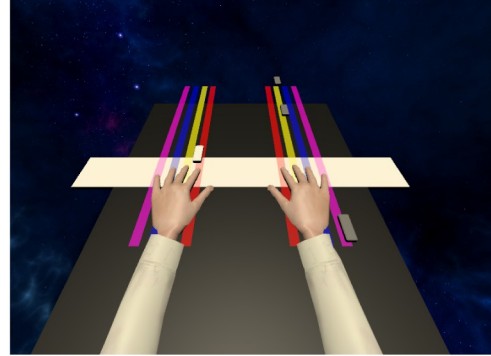

**Fig 1. Experimental context.** (a) Participants are comfortably seated in front of a table that is calibrated and replicated in the virtual environment. They are immersed in VR using an HMD. An active tracking solution with gloves was used to acquire fingers' motion in real-time. According to instructions, participants press on a foot pedal to report some specific events. (b) When participants do the task, they see height vertical colored lines on the virtual table: one per finger except for thumbs (four on the left and four on the right). Little white buttons are sliding down along the lines and eventually pass above a finger. The goal is to lift the corresponding finger to validate the buttons in the white area. When validated, the button disappears. In this illustration, the subject should be ready to lift the left index as the button is about to pass over it.

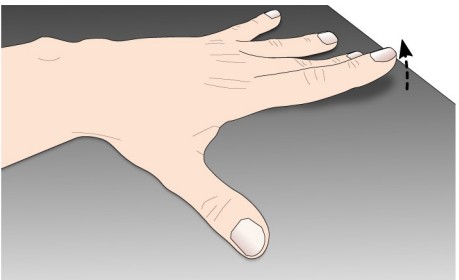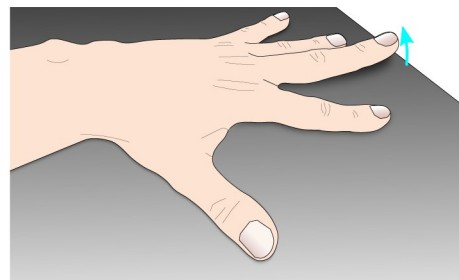

**Fig 2. Schematic illustration of a swap in fingers' motion.** The original motion of the index (i.e., the real motion from the user, the dashed arrow) is redirected onto the middle finger the user sees (i.e., the displayed motion, the cyan arrow). (a) Subject's real movement from the *real source finger*. (b) What the subject sees, i.e., the *displayed destination finger*.

## Task

A gamified finger-movement task was implemented in order to provide participants with a stimulating and challenging task, for which the level of difficulty can be adjusted to maintain an overall success with occasional spontaneous errors (SE). While playing the game, participants are also asked to perform a perception task, consisting in detecting if they noticed the animated finger was not the same as the finger they moved.

Participant's gaming task is to hit buttons that move downwards towards them as they pass over their fingers within the validation area (see Fig 1b). When successfully hitting a button, a short validation sound is played (0.235s) and the button disappears. The movement chosen is a lift to ensure that, except for the actuated finger, all the other fingers remain static owing to the table contact. The challenge of the task comes from the difficulty of following the activity on both sides (for the left and right hands) and for keeping the pace as the speed progressively augments during the game.

The speed of button is automatically and smoothly adjusted by the system to maintain the experience's flow and the difficulty of the task: When the subject performs correctly, the speed continuously increases so that at the highest speed, the subject cannot cope with the game's pace and makes mistakes (missing buttons or mixing fingers). Conversely, when the subject makes mistakes, the speed is reduced drastically to avoid overflowing participants.

Buttons are randomly distributed on each line, and the distance between those buttons remains consistent. The vertical lines laterally follow fingers' positions to ease the task by reducing the amount of attention required and the physical fatigue (movements are not physically guided like on a piano). No physical touch is simulated, and no scores are registered. The game ensures there is always only one button to hit at a time. Therefore, to prevent incidental multiple fingers lifts, the game is interrupted when more than one finger is actuated (followed by a reset of the table with buttons spawning from the beginning of the table).

The important specificity of this game for our experimental manipulation is that the machine will decide at some points to introduce finger swaps (Fig 2) and participants are asked to press the foot pedal when they detect such event. Subjects are informed that they have two seconds to react after they see a finger swap (during which the system cannot introduce more finger swaps) but are not specifically asked to press the pedal as fast as possible. During the experiment's tutorial, participants are trained to recognize such swaps (they must test finger swaps at least three times per hand). Once the pedal is pressed, the game immediately stops, and participants report their confidence level about their perception of the swap on a discrete scale within $[\![0, 10]\!]$. (N.B., To reduce bias while answering the question the scene is

made empty and the selection is done by maintaining the selection cursor, attached to the gaze direction, in one of the eleven values). Zero means 'I am not sure that the machine introduced a swap', and ten means 'I am sure the machine introduced a swap'. Once the value is validated, the questionnaire disappears, and the game restarts with new buttons at the top of the table.

## Implementation

### Speed regulation

To provide an environment maintaining the flow of the experience [27], the game's speed is continuously adjusted through a system inspired by proportional integral derivative controllers (PIDs) aiming to ensure a sufficient amount of SE (targeted value set in the algorithm loop: 10% of total amount of button press). Preliminary practice and speed assessment sessions are used to establish a reference speed for each user. Speed is contained between $0.35 m/s$ and $1.5 m/s$, and the acceleration is capped between $+0.05 m/s^2$ and $-0.15 m/s^2$ to avoid yanks and surprise effects. Those values were assessed with few pilots to ensure that the game remains engaging, not too demotivating and to ensure that the needed minimal count required for the analysis could be reached within a 1h long session for the subject.

**Automatic introduction of finger swaps.** To decide when a swap should be introduced, the system must first detect when a finger is moved (lifted) by the participant. A calibration process inspired by the work from Pavllo et al. [25] was used to store all fingers' vertical's position reference when in contact with the table. Then, an offsetted hysteresis filter continuously compares fingertips' vertical positions to the reference to detect which finger is moved. Before the filter raises this event, no finger swaps can be introduced. This is also used to detect SE and tell the speed regulator when the user made a mistake.

When a SE occurs (i.e., the moving finger is not on the button's line), the system randomly decides to trigger (or not) a correction of the movement. If the correction is triggered, the expected finger movement is swapped with the wrongly moved one. To distribute these error corrections over time, the algorithm enforces that every consecutive chunk of six decisions is balanced (i.e., it triggers randomly three among six cases of corrections and lets the three other movements uncorrected). Of note, pilots showed that six was enough to make sure participants could not predict a pattern. Maintaining a low chunk size ensures that conditions are continuously balanced instead of accumulating unbalance that would need to be fixed toward the end.

When the participant correctly hits buttons without SE, the system introduces five swaps per 100 trials. In such **EI** condition, the swapped finger is randomly chosen.

## Experimental design

The study was undertaken in accordance with the ethical standards as defined in the Declaration of Helsinki and was approved by our local Ethical commission. No minors were involved in this study, and consent was collected on written sheets before the beginning of the experiment. The protocol for this experiment is presented in Fig 3 and detailed in the following sections.

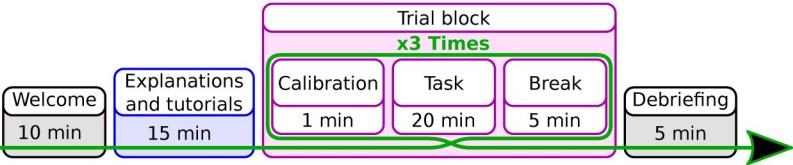

**Fig 3. Timeline illustration of the experimental protocol of the experiment.**

Participants are welcomed before giving their informed consent and filling out a demographic questionnaire and are equipped with tracking gloves (Fig 1a) and HMD. Once the experiment starts, the room's lights are turned off, and all the explanations are given in VR to ensure that all participants receive the same instructions.

## Explanations and tutorials

The first step explains the participants how to calibrate their hands to provide them with virtual hands during the tutorial. Then the game is explained to the participants and they can practice briefly (20 seconds). This is followed by a speed-assessment practice run during which the automatic speed adjustment of the game is monitored: the speed increases as the participant is successful (and reduces upon mistakes), enabling the system to store a personalized initial speed for each participant. The experimenter also observes participants' ability to do the task.

Participants then undergo a multitask-assessment practice run to ensure they can do both the game and the experimental tasks simultaneously. Here, the goal is to play the game and to press the pedal when a validated button turns red (500ms) instead of disappearing. Of note, the confidence question is introduced here to stagger instructions and ease comprehension, therefore, once the pedal is pressed, subjects give their confidence level in the observed presence of a red button (similar to the real task).

To ensure subjects understand the expected phenomenon to be reported, they are explained finger swaps and can try those by lifting fingers while the swap alternates between enable/disable to highlight the effect (with an indicator displayed in the scene). At this point, subjects are invited to ask questions to ensure all instructions are fully understood.

Finally, participants go through a dry-run to ensure that everything works and that the participant can perform the task correctly. At this point, a continuous white noise sound is added (to prevent the participant from hearing sounds from the real environment) and the participant is ready to perform the task for this study.

Of note, warnings are automatically raised and displayed to the user when multiple fingers are lifted simultaneously or when the subject moves his hand off the table. Also, the experimenter can trigger a message to stop the experiment in case of need.

## Trial block

During a trial block, participants perform the game task (hit buttons) and the experiment task (press pedal when detecting a swap) until 20 occurrences of each condition are reached (Fig 4). Each trial block is followed by a break when participants can remove the HMD, gloves and leave the chair before re-calibrating their hands for the next trial. On average, trial blocks lasted roughly 18min and presented 2788 buttons. Among those buttons, on average 38 are **EI**, and 50 are SE of which 31 are **EC**.

At the end of the experiment, feedback is considered, and participants receive monetary compensation for their time. The average session duration was designed to last 1h30 for roughly 3000 buttons presented.

## Hypothesis, measurements and analysis

### Formal hypothesis

The translation of the research question through the experimental setup can be formalized with the following hypotheses with the different conditions described in Fig 4.

The experimental conditions for our study are **EI** (error introduction) and **EC** (error correction). The **EI** condition represents the case where a participant moved the correct finger,

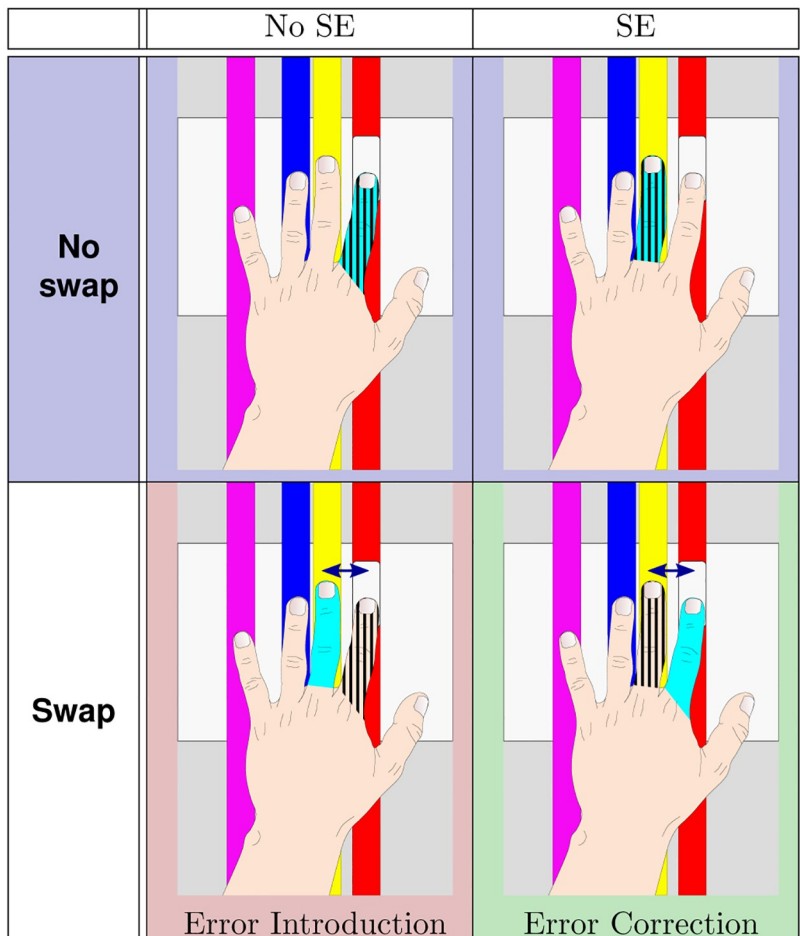

**Fig 4. Experimental conditions: Hatched areas indicate which finger is (sometimes wrongly) moved by subjects while cyan areas indicate the finger which is actually animated by the system.** Illustrations represent an example case when a subject should move the index to validate the button (in the game, buttons can arrive on any vertical line). The arrow in the swap conditions represents the swap count. Here its value is one as swapped fingers are next to each other.

and the system remapped this movement onto another finger, thus preventing the participant from succeeding. Conversely, the **EC** condition represents the case where the participant moved the wrong finger, and the system remapped this movement to the finger facing the button, thus helping the participant to succeed. Other conditions represent congruent visual feedback and are used to help maintain the game's flow.

The buttons' speed regulation is expected to push participants to make approximately 10% of SE over the total number of cases. The experimental system then introduces **EC** conditions for half of the detected SE cases and introduces **EI** conditions for 5% of the cases.

To elicit whether or not the motor conflict from finger-swaps [19] could lead to the self-attribution of finger-swapped actions in immersive VR, and to assess if the direction of the distortion (i.e., hindering or helping) at achieving a goal-oriented task [12, 20] affects the former self-attribution, we formulated the following hypothesis:

**H1—Introducing a swap in fingers' motion to prevent users from reaching the goal is more rejected than a swap helping to reach the goal**.

As Henmon et al. showed that high confidence levels are correlated with faster reaction times [28], our second hypothesis, expecting a higher confidence level at detecting penalizing finger swap compared to the ones helping the user, was extended with a shorter reaction time for the former condition, and therefore formulated as:

**H2a.—Introducing a swap penalizing the user is reported with a higher confidence level than a swap helping the user**

**H2b.—Introducing a swap penalizing the user is reported with a shorter reaction time than a swap helping the user**.

## Measurements

As per the instructions, participants press the foot pedal when they observe a finger swap. In such an event, the game immediately stops, and the system stores the pedal press time, allowing to measure (post-analysis) the amount of time elapsed since the previous experimental condition (**EI** or **EC**).

Of note, the condition is ignored for further analysis if the participant did another mistake in the time between the introduction of the experimental condition and the pedal press. This applies for SE as well as for conditions followed with a warning (e.g., multiple fingers moved simultaneously).

In addition, the amount of swap (swp), referring to the number of hops between the finger moved and the animated one, is also stored for post-analysis (a swap count of 0 means there is no swap, a swap count of 1 means that the swapped finger is the direct neighbor of the moving finger such as on Fig 2, and so on).

Finally, after a pedal press, a questionnaire asks the participant its confidence in the observation of a finger swap.

## Statistical analysis

Scripts and data used in the analysis are available in the S1 Dataset.

As our hypotheses only concern **EI** and **EC** conditions the dataset was filtered to remove other conditions (no mistake without error introduction and mistakes without error correction). The experimental design considers that a pedal pressed under no swap condition with no spontaneous error is assumed to be attached to the previous experimental condition unless the time window is closed. The analysis was conducted using ®.

We expect that introducing a swap in fingers' motion to prevent users from reaching the goal will be more rejected than a swap helping one to reach the goal. This can be formalized as Eq 1.

$$\mathbb{P}(\texttt{pedal\_pressed}|\,\mathbf{EC}) < \mathbb{P}(\texttt{pedal\_pressed}|\,\mathbf{EI}) \tag{1}$$

Therefore, a mixed model providing the pedal pressed outcome (pp) as a factor of the fixed effects of the amount of swap (swp) and SE (se) was used to fit our filtered dataset (3256 points) to assess this hypothesis. The logit function, defined as $\texttt{logit}: x \mapsto \ln\left(\frac{x}{1-x}\right)$, was used as the outcome pp is binary (the pedal is either pressed or not within the two second time window after a condition). Plots observations hinted at a square factor for the swp predictor and did not highlight interaction factors. Therefore we considered the outcome as a mean (*I*)

plus the impact of se, swp and swp$^2$ (Eq 2).

$$\mathbb{P}(\texttt{pp}) = \texttt{logit}(I + \beta_1 \cdot \texttt{se} + \beta_2 \cdot \texttt{swp} + \beta_3 \cdot \texttt{swp}^2) \tag{2}$$

Random effects from the subject id, the elapsed time since the last condition (to relate the speed of actions), and the number of minutes elapsed since the experiment beginning (to relate fatigue) were also considered (although not displayed in the formula).

The model's fitness was assessed using the residual analysis using ®'s DHARMa package with a 0.18 *p*-value for the KS test of deviation, 0.66 for the dispersion test, and 0.77 for the outliers test.

We expect that introducing a swap penalizing the user would be reported with a higher confidence level than a swap helping the user through a higher confidence level, and a shorter reaction time. This is equivalent to Eqs 3 and 4.

$$\mathbb{E}(\texttt{confidence\_level}|\mathbf{EC}) < \mathbb{E}(\texttt{confidence\_level}|\mathbf{EI}) \tag{3}$$

$$\mathbb{E}(\texttt{response\_time}|\mathbf{EI}) < \mathbb{E}(\texttt{response\_time}|\mathbf{EC}) \tag{4}$$

Only conditions with a pedal pressed were retained (783 entries) as the confidence level questionnaire and the reaction time (time elapsed between the button's validation and the pedal press) are only defined after a pedal pressed.

A two-sided two-samples median permutation test (with 50 000 iterations) was used to compare confidence levels and reaction time medians between both groups (**EI** and **EC**) followed with the one-sided test to retrieve the direction of the difference.

## Results

### Demographics

20 participants aged between 18 and 44 years old (median: 24, std: 5.62), including 10 women, participated in this experiment. One participant stopped the session due to the difficulty of wearing the HMD. Most participants came from *anonymous* area and were all students or people working in academics. One participant was left-handed. The demographic questionnaire indicates that participants mostly experienced VR a few times, were healthy and comfortable with typing.

Over 52,977 buttons pressed 3,256 entries were retained to assess the first hypothesis (sum of all swap conditions, in red and green in Table 1). The assessment of the second hypothesis used the subset of the filtered dataset where the pedal was pressed, subset composed of 783 entries (sum of all swap conditions where pp $\neq$ 0).

**Table 1. Detailed count of each case occurrence.** The No Swap cases (blue) do not count for the experimental condition evaluation as they represent the large majority of 'normal' events when playing the game. The pp columns represent the sub pedal pressed count per condition while the% pp represents its percentage share. In total, on the experimental conditions, 24% of the swaps were noticed with a pedal press (sum of pp over totals from red and pp over totals from green cells from the last line).

| | swp | No SE ($\simeq$ 95%) | | | SE ($\simeq$ 5%) | | |
|---|---|---|---|---|---|---|---|
| | | Total | pp | %pp | Total | pp | %pp |
| **No swap** | 0 | 47955 | 20 | 0.04% | 1766 | 238 | 13.48% |
| **Swap** | 1 | 1187 | 370 | 31.17% | 1021 | 44 | 4.31% |
| | 2 | 567 | 215 | 37.91% | 35 | 2 | 5.71% |
| | 3 | 442 | 152 | 34.39% | 4 | 0 | 0% |
| | Total | 2196 | 737 | 33.6% | 1060 | 46 | 1.63% |

**Table 2. Predictors values for the fitted mixed model.** We can observe the very significant impact of the `se` predictor on the observed outcome.

| Fixed effect | Equation factor | Estimate | *p*-value |
|:---:|:---:|:---:|:---:|
| Intercept | $I$ | −1.72 | $2.54 \cdot 10^{-05}$ |
| `se` | $\beta_1$ | **− 2.37** | $< 2 \cdot 10^{-16}$ |
| `swp` | $\beta_2$ | 0.94 | 0.0322 |
| I(`swp`$^2$) | $\beta_3$ | −0.28 | 0.0536 |

## Finger swap detection

Fitted coefficients for the model (defined in Eq 2) are displayed in Table 2.

The *p*-value associated with the `se` predictor coefficient is low ($p < 2 \cdot 10^{-16}$); thus, the odds of the observed effect from this predictor being due to chance are almost null. Since the `se` predictor coefficient is negative (and the `logit` transform function is an increasing function), the odds of pedal pressed are significantly lower when `se` = 1 (**EC**) compared to when `se` = 0 (**EI**) given the model used (Eq 2). This translates into the red curve of **EI** being above the green one of **EC** in Fig 5a. Finally, the $R^2$ was measured at 0.21 (interpreted as low [29]) for the whole model, and at 0.16 for the only effect size from the `se` factor (also interpreted as low). Therefore, introducing a swap in fingers' motion to prevent users from reaching the goal will be more rejected than a swap helping one to reach the goal, hence validating our first hypothesis.

Additionally, we observed that the amount of swap (i.e., `swp`) also significantly impacts the odds of having a pedal pressed. A possible explanation could be that swaps of neighboring fingers are harder to observe than those from distant fingers (e.g., index and pinky).

## Confidence level and reaction time analysis

A significant difference was measured between the two samples' median through the two-sided test (a 0.001 *p*-value). Cohen's D effect size for the raw influence of the self error on the

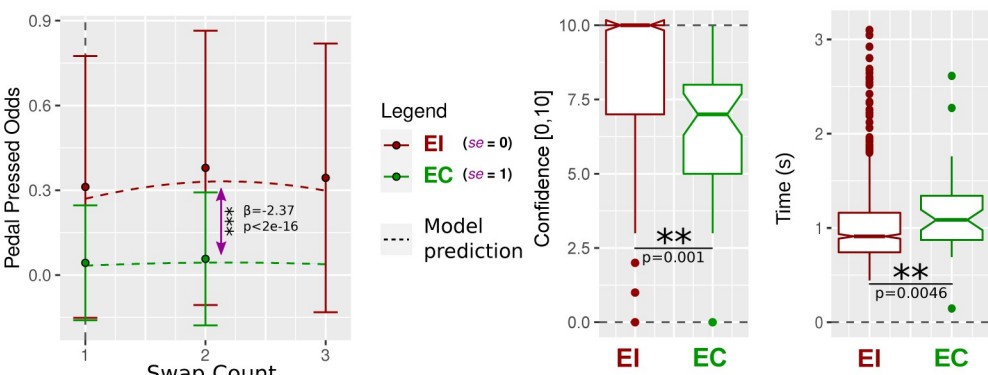

**Fig 5. Plots of the results from the analysis section.** (a) H1—Probability of pressing the pedal after the introduction of a finger swap, for swap count of 1 (e.g., swap index finger with middle finger), 2 (e.g., index finger with ring finger), and 3 (index finger with little finger, this occurred only for the **EI** condition, i.e., there is no data-point for **EC**). Plots of the mixed model prediction (dashed lines) compared to all subjects' data. The pink arrow highlights the significant difference resulting from the se fixed effect. (b) H2a.—Effects of **EI** and **EC** conditions in terms of confidence levels. The **EI** condition presents a very significantly higher confidence score than the EC condition. (c) H2b.—Effects of **EI** and **EC** conditions in terms of reaction times. The **EI** condition presents a very significantly shorter reaction time than the **EC** condition. (Two additional outliers at 9.1s and 3.9s are out of the plot for **EI**).

confidence level was measured at 0.87 (high). The confidence level median for the **EI** is 10 with a mean of 8.39 and a standard deviation of 2.19. In comparison, for the **EC**, the median is at 7, the mean is 6.46 and the standard deviation is 2.66. This difference is oriented with a higher median for the **EI** condition compared to the **EC** condition (a 0.001 *p*-value). Those results are plotted in Fig 5b and validate the first part of our second hypothesis (H2a., Eq 3).

The same procedure yield a 0.00456 *p*-value for the two-sided test, revealing a significant difference between both samples' medians, with a shorter reaction time for the **EI** condition compared to the **EC** (0.0042 *p*-value, Fig 5c) which validates H2b. (Eq 4). Cohen's D effect size for the raw influence of the self error on the reaction time was measured at 0.24 (low). The reaction time median for the **EI** is 0.91s with a mean of 1.02s and a standard deviation of 0.51s. In comparison, for the **EC**, the median is at 1.09, the mean is 1.15s and the standard deviation is 0.41s.

It is to be noted that in the **EI** condition many outliers were observed. A possible explanation might be the fact that in case of doubts people might take a bit longer to decide whether a swap was introduced or not and in such a case, they would more easily recognize an error introduction and press the pedal while in the other case, they might just accept it.

This does not contradict the relation from [28] and, together, those results support our second hypothesis.

## Discussion

Based on prior results from the literature [6, 20], we expected that introducing finger swaps in an engaging game would induce a different behavior when the swap helps rather than hinders participants in their task. More specifically, we expected lower odds of perceiving the swaps when helping compared to when hindering the user. Additionally, we expected that the latter case would be detected with higher confidence. This study was designed to answer those two questions by providing subjects with an engaging task performed in immersive VR, with an avatar following users' movements. Confidence levels and reaction times were measured to assess the subject's confidence in the reported swap.

Results validate our two hypotheses: participants are less sensitive with lower confidence levels and lower odds of detecting when swaps help them than when they hinder their movements.

Up to an offset of ∼55%, we observed the same detection rate of an alteration of the user's action as in the work from Logan et al. [12]. In their study investigating the self-attribution of corrections/inserted errors, authors reported a detection rate of altered actions of approximately 85% for an inserted error against 55% for a corrected error. In comparison, in our study, the values observed were 33% for error introduction to the 1.63% in the error correction, and a similar drop of ∼30% was measured in the perception rate between those two conditions.

Since the discrepancy we introduce in both cases (i.e., a finger swap with the same algorithm on the same game) is the same, a purely sensory-motor comparison with the visual feedback should raise the same warning for any type of swap introduced. Therefore, the comparator model for self-attribution of movement [7] does not fully explain the observed behavior. Our results rather corroborate the work of Logan et al. [12], showing that the authorship illusion is composed of at least two stages (referred to as the inner and outer loop), and consistent with a hierarchical error-detection mechanism.

### Relation with agency and embodiment

Although the levels of SoA and SoE were not evaluated during this experiment, the experimental setup with its immersive technology, the self-location of the virtual avatar, and the

animation of upper limbs was assumed to provide users with relatively good levels of embodiment and agency, at least comparable to those in similar experiments [19, 22]. Conversely, a pedal press at the detection of a finger swap thus indicates a disruption of the user's SoA (and probably of SoE).

Results can therefore be interpreted in terms of agency in the following way: finger-swaps in the **EI** condition are more likely to disrupt SoA than in the **EC** condition. Furthermore, considering the low probability of detection of **EC**, our results suggest that error correction with finger swap has a limited impact on the SoA.

In practice, to avoid disrupting the SoA, it is important to prevent users from noticing finger swaps, and **EI** should be avoided. A system can introduce swaps in finger motion in immersive VR in order to help participants achieve a task without them noticing (most of the time) and with a limited impact on SoA and SoE. Such results can be useful for controlling the flow in a training task and maintaining motivation (e.g., learning the piano, typing) or for compensating for finger tracking errors (i.e., in the absence of correct tracking, trigger the expected finger movement).

## Limitations and future works

Our experimental manipulation required to place participants in a situation leading to spontaneous errors. Other approaches, with a question following each trial (as in the work from Salomon et al. [19] or Balslev et al. [30]), could not be used as participant would have constantly been interrupted, breaking the flow of the game [27]. Instead, using a method similar to the one from Kokkinara et al.'s study [31], we asked subjects to self-report the introduction of finger swaps through a pedal press. Our design can thus only assess perceived swaps, and cannot reveal behaviors based on non-observed finger swaps. It is indeed possible that participants deeply engaged in the game might have forgotten to report some swaps, or that their attention might have been temporarily disrupted. Using eye-tracking might help reveal some unexpected behaviors and/or disentangle some conflicting cases (e.g., measuring eye saccades when a swap occurs or not). However, current HMDs do not provide fast enough eye-tracking capabilities to measure those saccades (requiring a sampling frequency to be above 500Hz) [32], and knowing participants' gazes is not necessarily sufficient to relate the actual perception of the change by the user (e.g., movements can very well be perceived in peripheral vision).

Another differentiation with traditional approaches is that the participant's attention is shared between two tasks (the game with validating buttons and reporting finger swaps). Although all participants underwent multitasking training and assessment sessions, it remains unknown how much our results are influenced by their ability to perform the dual-task for the specific experimental manipulation. Further testing and evaluation of participants could be conducted to achieve a more detailed understanding of the interactions between the ability for multitasking and the experience of embodiment.

Compared to the study from Burns et al. [15] the subject's task is more complex in our case; hence, all participants had to be trained on the type of distortion to recognize (finger swaps). As a consequence, given that warning the participants about distortions influences the experiment outcome [12, 15], it is normal to have higher odds of swap notifications in our context. It could be expected that, without previously informing participants of the possibility of finger swap, the detection rate would be much lower. To study such cases, a system monitoring brain activity with electroencephalography could be used to monitor the brain's spontaneous reactions to error, known as Error Related Potential [33, 34]. As previously done for detecting violations of agency in VR [35, 36], it would probably be possible to directly detect the brain reaction to error correction or error introduction, without interrupting the participant, and

with the possibility to answer to mechanistic and neurological questions on the agency of error correction.

## Conclusion

Our study shows that virtual distortions of finger movements (swaps) that help users to reach a target with fingers are more tolerated than distortions hindering their action. This extends the previously observed effect of distortions for full arm reaching tasks in VR [6, 20], thus generalizing the observation that some carefully designed discrepancies between real and virtual body movements can be well tolerated as far as they help in achieving a goal in VR.

More specifically, our experimental setup successfully elicited the self-attribution of finger-swaps in immersive VR, with a significant difference between swaps helping or not the subject to accomplish a challenging task. Our results support the hierarchical error-detection mechanism proposed by Logan et al. [12], with inner loops taking care of the details of performance (here finger swaps) and outer loops ensuring that intentions are fulfilled, thus leading to the authorship illusion for avatar-corrected actions.

Eventually, one take-home message for designers of embodied interaction in VR involving finger movements is that a system can introduce finger swaps without disrupting the SoA as long as those swaps help users in achieving the task at hand.

## Supporting information

**S1 Appendix. Appendix.** An appendix detailing the demographic questionnaire, the used inverse kinematics, and the finger swap animation is provided as a separate document.
(PDF)

**S1 Video.**
(WEBM)

**S1 Dataset.**
(ZIP)

## Author Contributions

**Conceptualization:** Mathias Delahaye, Ronan Boulic, Bruno Herbelin.

**Data curation:** Mathias Delahaye.

**Formal analysis:** Mathias Delahaye, Bruno Herbelin.

**Funding acquisition:** Ronan Boulic, Bruno Herbelin.

**Investigation:** Mathias Delahaye.

**Methodology:** Mathias Delahaye, Ronan Boulic, Bruno Herbelin.

**Project administration:** Ronan Boulic, Bruno Herbelin.

**Resources:** Ronan Boulic.

**Software:** Mathias Delahaye.

**Supervision:** Olaf Blanke, Ronan Boulic, Bruno Herbelin.

**Validation:** Mathias Delahaye.

**Visualization:** Mathias Delahaye.

**Writing – original draft:** Mathias Delahaye.

**Writing – review & editing:** Mathias Delahaye, Olaf Blanke, Ronan Boulic, Bruno Herbelin.

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
