## [Decision Letter · Decision Letter 0]

7 Jun 2022

PONE-D-22-07695Avatar error in your favor: Embodied avatars can fix users’ mistakes without them noticingPLOS ONE

Dear Dr. DELAHAYE,

Thank you for submitting your manuscript to PLOS ONE. After careful consideration, we feel that it has merit but does not fully meet PLOS ONE’s publication criteria as it currently stands. Therefore, we invite you to submit a revised version of the manuscript that addresses the points raised during the review process.

We look forward to receiving your revised manuscript.

Kind regards,

Markus Lappe

Academic Editor

PLOS ONE

Journal Requirements:

2. Please provide additional details regarding participant consent. In the ethics statement in the Methods and online submission information, please ensure that you have specified what type you obtained (for instance, written or verbal, and if verbal, how it was documented and witnessed). If your study included minors, state whether you obtained consent from parents or guardians. If the need for consent was waived by the ethics committee, please include this information

3. We note that you have stated that you will provide repository information for your data at acceptance. Should your manuscript be accepted for publication, we will hold it until you provide the relevant accession numbers or DOIs necessary to access your data. If you wish to make changes to your Data Availability statement, please describe these changes in your cover letter and we will update your Data Availability statement to reflect the information you provide

Reviewers' comments:

Reviewer's Responses to Questions

**Comments to the Author**

1. Is the manuscript technically sound, and do the data support the conclusions?

Reviewer #1: Yes

Reviewer #2: Yes

2. Has the statistical analysis been performed appropriately and rigorously? 

Reviewer #1: I Don't Know

Reviewer #2: Yes

3. Have the authors made all data underlying the findings in their manuscript fully available?

Reviewer #1: Yes

Reviewer #2: Yes

4. Is the manuscript presented in an intelligible fashion and written in standard English?

Reviewer #1: Yes

Reviewer #2: Yes

5. Review Comments to the Author

Reviewer #1: First of all, I really liked the paper and the experiment and I thought the approach was really interesting and innovative. But I have some questions I would want to have answered/explained in the mansucript, especially regarding the results section, that is why I recommendated a revision of the paper.

Abstract:

It would be nice to find the 24% of noticed swaps somewhere in table 2 again.

Introduction:

I think it is technical sound and gives a nice overview over the topic and the relevant literature. I think it's focus might be a bit to much on sense of agency, since this is not really measured in the study itself. Maybe it makes sense to include a section about the rubber hand illusion, redirected pointing (thresholds) and early prism adaptation studies including pointing? Since these works are more about the perception of visual hand manipulation.

Setup:

technical sound and allows to replicate the experiment. I was wondering since the pedal is a midi padel: What midi value did you use for the pedal-is-pressed variable? (midi pedals typically encode a linear scale from 0 to 127 which represents the pressing state)

Why did you chose chunks of 6 for balancing?

I am not sure if you reached your target of ~10% of SE. All in all I think another table row in table 2 showing the sums of all swp conditions would make it easier to quickly get an overivew.

Why did you add white noise? I guess this is about white noise sound? and nothing visual (at least I did not notice anything in the video)?

Hypothesis Measurement and Analysis

Are 50 and 25 the real average number of SE, EI and EC per block or just rounded estimates?

I would suggest to split H2 into H2.1 confidence level and H2.2 reaction time.

313 to measureS

warnings should be described earlier e.g. in explanation and tutorials. What other warnings existed?

Statistical Analysis:

I think this section would benefit from being structured along the hypotheses. Is the logit model done to also test H2? or is this is a more exploratory analysis? I was also wondering if it makes sense to include both a swp and swp^2 in the same model since the will naturally be highly correlated especially when using a variable like swp (on a scale from 0 to 3)? Therefore the two factors are likely to explain the same variance in the data? I guess it would be useful to test a model with against a model without the linear factor?

Fig 4a: Is this one subject? or the average of all subjects?

Although I think the median permuation test presented here, might be a legitimate approach I think it isn't the standard way of analyzing reaction time data. So please explain in detail why you decided to use this method instead of maybe a mean based approach on log(rt). Are your results only visible in your median based approach?

Moreover I don't understand why you used a median based (instead of mean based) approach for the confidence ratings, since these were answered on a likert-like scale? Please discuss!

In Fig 4 c we can see a lot of outliers in the left boxplot, please discuss!

The most important task:

Please also explicitly report the effect sizes and/or the mean/median values (including sds) of each condition instead of only presenting the p-values and figures.

Discussion:

Relation with Agency and Embodiment

I think especially the second paragraph is quite notional.

I was wondering if your results are comparable to the user experience of auto-correct-software during typing. I am not an expert in this field, but maybe it makes sense to discuss this?

Limitations

How in detail can eye tracking and more testing help? Why did you not do this in this study?

Dataset:

on which scale are sportiness and vr experience? (min, max?)

Reviewer #2: The study of Delahaye and co-workers investigated whether a finger swap helping users to successfully complete a task is more tolerated with respect to another condition where the finger swap penalized them within a virtual reality environment. Specifically, the task consisted in validating buttons by the fingers visually projected into the VR and how participants detected anatomical swaps with and without spontaneous errors (SE). In the context without SE, the authors assessed the condition of error introduction (EI), whereas, in the with SE context, they assessed the condition of error correction (EC). While performing the task, participants were asked to perform a perception task, consisting in detecting if they noticed the animated finger was not the same as the finger the moved.

This is an interesting study, so I have only a minor comment that can improve the clarity of the work.

Minor:

Despite a detailed explanation of the task and the setup used, I do not understand results in table 3 and how they demonstrated the first hypothesis of the study. Please add more text in order to better explain this point.

6. PLOS authors have the option to publish the peer review history of their article (what does this mean?). If published, this will include your full peer review and any attached files.

Reviewer #1: **Yes: **Niklas Stein

Reviewer #2: No

---

## [Decision Letter · Decision Letter 1]

16 Aug 2022

Avatar error in your favor: Embodied avatars can fix users’ mistakes without them noticing

PONE-D-22-07695R1

Dear Dr. DELAHAYE,

We’re pleased to inform you that your manuscript has been judged scientifically suitable for publication and will be formally accepted for publication once it meets all outstanding technical requirements.

Kind regards,

Markus Lappe

Academic Editor

PLOS ONE

Additional Editor Comments (optional):

Reviewers' comments:

Reviewer's Responses to Questions

**Comments to the Author**

1. If the authors have adequately addressed your comments raised in a previous round of review and you feel that this manuscript is now acceptable for publication, you may indicate that here to bypass the “Comments to the Author” section, enter your conflict of interest statement in the “Confidential to Editor” section, and submit your "Accept" recommendation.

Reviewer #1: All comments have been addressed

2. Is the manuscript technically sound, and do the data support the conclusions?

Reviewer #1: Yes

3. Has the statistical analysis been performed appropriately and rigorously? 

Reviewer #1: Yes

4. Have the authors made all data underlying the findings in their manuscript fully available?

Reviewer #1: Yes

5. Is the manuscript presented in an intelligible fashion and written in standard English?

Reviewer #1: Yes

6. Review Comments to the Author

Reviewer #1: I think the paper should be accepted.

Anyway, here is some minor stuff (mostly language):

p7: Here, a within-subject experiment was condutected to determine wether a finger swap helping users to achieve a task would be more tolerated than one penalizing them.

p8: feedback is considered

p12: (timout)?

p12: system inspired by proportional integral derivative controllers (PIDs) aiming...

p12: demotivating and to ensure that the cases minimal count required for //"needed" instead of "cases"?

p13: Then the game is explained to the participants and they can practice briefly (20 seconds).

p15: To elict (-s) whether or not the motor conflict from finger-swaps...

p18: and press the pedal while in the other case

p19: Since, the discrepancy we introduce in.....

p19: However, current HMDs do not provide fast enough eye-tracking capabilities to measure those saccades (requiring a sampling frequency to be above 500Hz), <-- yes indeed: Stein, N., Niehorster, D. C., Watson, T., Steinicke, F., Rifai, K., Wahl, S., & Lappe, M. (2021). A Comparison of Eye Tracking Latencies Among Several Commercial Head-Mounted Displays. I-Perception. https://doi.org/10.1177/2041669520983338 :)

Fig 4a: is the red dot above the green dot? Maybe small jitter would be helpful?

Fig 3: I would suggest to not use green text/ a green arrow on red background

7. PLOS authors have the option to publish the peer review history of their article (what does this mean?). If published, this will include your full peer review and any attached files.

Reviewer #1: **Yes: **Niklas Stein

---

## [Editor Report · Acceptance letter]

8 Jan 2023

PONE-D-22-07695R1 

Avatar error in your favor: Embodied avatars can fix users’ mistakes without them noticing 

Dear Dr. Delahaye:

I'm pleased to inform you that your manuscript has been deemed suitable for publication in PLOS ONE. Congratulations! Your manuscript is now with our production department. 

Kind regards, 

on behalf of

Dr. Markus Lappe 

Academic Editor

PLOS ONE